# Parental Views on the Acceptability and Feasibility of Measurement Tools Used to Assess Movement Behaviour of Pre-School Children: A Qualitative Study

**DOI:** 10.3390/ijerph19063733

**Published:** 2022-03-21

**Authors:** Sophie M. Phillips, Carolyn Summerbell, Kathryn R. Hesketh, Sonia Saxena, Frances C. Hillier-Brown

**Affiliations:** 1Department of Sport and Exercise Sciences, Durham University, Durham DH1 3HN, UK; carolyn.summerbell@durham.ac.uk; 2The Centre for Translational Research in Public Health (Fuse), Newcastle University, Newcastle upon Tyne NE1 7RU, UK; frances.hillier-brown@newcastle.ac.uk; 3MRC Epidemiology Unit, University of Cambridge, Cambridge CB2 0QQ, UK; krh40@medschl.cam.ac.uk; 4Population Policy & Practice Research and Teaching Department, UCL Great Ormond Street Institute of Child Health, London WC1N 1EH, UK; 5School of Public Health, Imperial College London, London W6 8RP, UK; s.saxena@imperial.ac.uk; 6Population Health Sciences Institute, Newcastle University, Newcastle upon Tyne NE2 4AX, UK; 7Human Nutrition Research Centre, Newcastle University, Newcastle upon Tyne NE2 4HH, UK; 8Newcastle University Centre of Research Excellence in Healthier Lives, Newcastle University, Newcastle upon Tyne NE1 7RU, UK

**Keywords:** feasibility, acceptability, movement behaviours, measurement, qualitative research, pre-school children

## Abstract

Movement behaviours (physical activity, sedentary behaviour, and sleep) are important for the health and development of pre-school children (aged 3–4 years). There is limited qualitative research examining the acceptability and feasibility of tools used to assess movement behaviours in pre-schoolers. This study explored parental views on various measurement tools in three deprived areas in England, UK (West Yorkshire, County Durham and Northumberland). The study consisted of a demonstration of the different tools (accelerometers, a diary and a questionnaire), directly followed by focus group discussions. Three focus group discussions with a total of eleven parents and carers were transcribed verbatim and analysed using thematic analysis. Findings revealed four main themes: (1) importance of contextual information when using any measurement tool (e.g., child illness, capturing different routines); (2) practical issues associated with devices (e.g., aversion to devices being attached directly to the skin of their child; concern of larger devices during sleep time); (3) encouraging children to wear a device (e.g., making devices attractive to children*—*‘superpowers’); and (4) presentation of diaries and questionnaires (e.g., age-appropriate movement activities, preference for real-time recording over recall). Practical recommendations for the use of the tools to measure movement behaviours of pre-school children are provided.

## 1. Introduction

A recent paradigm shift suggests that the movement behaviours of physical activity, sedentary behaviour and sleep should be viewed collectively rather than in isolation [1,2,3,4] as time spent in one behaviour may modify the health-related influence of time spent in any other of the behaviours [5,6,7]. In 2019, the World Health Organization (WHO) published 24 h movement behaviour guidelines for children in their early years (0–5 years old) [8]. International guidance from several countries has also shifted the focus to integrated movement behaviour guidelines for children in their early years [9,10,11,12,13]. The WHO guidelines recommend that pre-school children (3–4 years old) should engage in at least three hours of a variety of physical activities per day, of which one hour should be moderate to vigorous physical activity. Children should not be restrained (e.g., strapped in a car seat or pram) for more than one hour at a time, or sit for extended periods, and sedentary screen time should be no more than one hour per day. Pre-school children should have 10–13 h of good quality sleep [8].

The measurement tool of choice is one of the first decisions in any research examining movement behaviours. Practical aspects to consider when measuring movement behaviours of any population include what the outcomes of interest are: scale of the research, cost of tools, participant and researcher burden, and recall and reactivity bias [14]. Although the key measurement properties (validity and reliability) are fundamental when selecting a measurement tool, the acceptability (whether individuals are willing to do something) and feasibility (whether individuals are able to do something) of a tool may determine success of the study [15,16]. The measurement tool of choice can affect retention in research studies and the amount of usable data, including through low compliance with accelerometer wearing protocols [17,18,19], with some research suspending the use of accelerometers during data collection for these reasons [20,21]. For example, the SUNRISE study, an international project using multiple measurement tools to collect data on movement behaviours of pre-school children, has suspended use of an accelerometer and modified existing parental reported tools due to feasibility and acceptability concerns [21].

There are currently no practical recommendations for the measurement of movement behaviours of pre-school children and limited research exploring the acceptability and feasibility of tools used for this purpose [22,23]. Studies examining acceptability have primarily used survey based research, whereby parents or researchers rate sentences such as ‘*the device was uncomfortable to wear*’ [24,25,26,27,28] or through researcher observation [29]. Whilst these studies are informative in inferring the practical acceptability of devices, there is a need for more qualitative research to understand how different tools are perceived and accepted in the target population to ensure they are appropriate and accessible prior to use [30,31,32,33]. This is particularly key to ensure that tools are equally useful across all levels of deprivation, and that research will not increase inequalities in evidence through participation in research being with more advantaged population samples only [34,35].

There are specific challenges associated with measuring movement behaviours of pre-school children, including the size and weight of device-based tools and increased likelihood of measuring equipment being played with [36]. Cognitive capabilities prevent young children from performing detailed recall of their own movement behaviours, resulting in the need for proxy (parental/carer) reports [37,38]. A greater understanding of the acceptability and feasibility of possible measurement tools is needed to further develop the evidence base and understanding of the movement behaviours of pre-school children [39]. As the importance of concurrently assessing these behaviours grows, there is a need to address measurement issues and practical considerations by looking at how various tools will be applicable for 24 h measurement of behaviours [5,40,41].

We have found only one study which conducted focus groups with mothers of toddlers (aged 2–3 years) to determine the qualitative feasibility of accelerometers used to assess physical activity of toddlers and their parents. This study identified issues that had potential to impact recruitment and compliance within their subsequent studies, and helped to select the accelerometer of choice in their future study [42]. At present, no study has qualitatively explored the acceptability and feasibility of a range of measurement tools used to assess the movement behaviours (physical activity, sedentary behaviour, and sleep) of pre-school children (aged 3–4 years).

The aim of this qualitative study was to examine acceptability and feasibility of a range of measurement tools used to assess the movement behaviours of pre-school children, in parents and carers living in areas of high deprivation in the North of England.

## 2. Materials and Methods

We obtained ethical approval from the Department of Sport and Exercise Sciences ethics committee at University of Durham, UK. The project was covered by Durham University’s public liability insurance and professional indemnity insurance. Participants’ anonymity was protected using participant numbers and pseudonyms.

### 2.1. Participants

Early years settings (children’s centres, nurseries and schools with pre-school provision) in the North of England were initially contacted using purposive sampling. All early years settings (hereon termed ‘*settings*’) contacted were in the highest quartile of index of multiple deprivations, indicating they were in the most deprived areas in the United Kingdom (UK) [43]. The settings were approached by telephone, e-mail or via personal network of the researchers. If a setting agreed to participate, an opportunity sampling method was used whereby posters and information sheets were provided to parents and carers of children at the setting. Participants were eligible to participate if they were a parent or carer of a pre-school child (aged 3–4 years old). Children were able to be present during the whole session (demonstrations and focus groups) to ensure inclusivity.

### 2.2. Data Collection

The study consisted of a demonstration of the different tools to give context and provide a focus to stimulate discussion, directly followed by a focus group discussion. All sessions took place between December 2019 and January 2020 at the participating settings in three deprived areas in England, UK (West Yorkshire, County Durham and Northumberland). The setting managers conversed with parents and carers to arrange an appropriate date and time for the focus group, based on when a group of interested potential participants were available. Participating parents and carers provided written informed consent at the beginning of the group session. They then completed a brief demographic information questionnaire, which included questions on: age, sex, ethnicity, age of child, education level, employment status, household income and home postcode. Following this, the main session took place. First, a demonstration and explanation of the measurement tools used to examine movement behaviours of pre-school children was delivered by the lead researcher (SMP). This included: (1) Accelerometers (Actigraph GT3X+ (ActiGraph, Pensacola, FL, USA), activPAL4 micro (PAL Technologies Ltd., Glasgow, UK) and Actical (Philips Respironics Inc., Murrysville, PA, USA)) (see Appendix A for images and detailed information of the devices), with accompanying instructions (see Appendix A) and logs and (2) proxy reported tools, including: a sleep diary and a questionnaire. The accelerometers were selected based on a review of the literature determining these to be most valid for this age group [22]. The sleep diary devised for the present study was a modified version of the National Sleep Foundation consensus diary [44]. The questionnaire was a provisional questionnaire that was being developed as part of a wider piece of research. Parents, carers and children had the opportunity to trial the measurement tools during the demonstration session.

The lead researcher (SMP) then conducted the focus groups using a semi-structured discussion guide. The discussions started by asking for initial thoughts and opinions on the tools, which stimulated discussion on the practicality, acceptability, suitability and any envisaged problems for using the methods to assess the behaviours of their pre-school child. Specific questions were asked regarding whether the tools and accompanying instructions were understandable, comprehensive and relevant, if discussion had not already covered this. Parents and carers were asked for their opinions on the use of each of the measurement tools for their child aged 3 or 4 years old, for a period of 7 days, as a 24 h wear protocol if they were to take part in a future research study. The choice of 7 days was based on this being the optimal and usual length of measurement [36,45]. The 24 h wear period was in order to assess the whole 24 h of the day [46], as well as higher compliance reported for 24 h wear protocols [47].

### 2.3. Data Analysis

All focus group discussions were voice recorded and transcribed verbatim. We used a thematic analysis approach to identify important themes and analyse patterns within the data [48,49,50,51]. We used an inductive approach in our analysis to ensure that all parental views on the tools could be captured, rather than within prescribed a-priori defined categories used in previous work [42]. The thematic analysis was conducted manually by the lead researcher (SMP) using a systematic step-by-step process proposed by Braun and Clarke [48] (see Appendix A for full details).

## 3. Results

Three focus groups with a total of 11 parents and carers (including pre-school teachers) of pre-school children (aged 3–4 years) took place. Children were present in two of the three focus groups. Of the nine early years settings contacted, a total of five settings were willing to participate, with three remaining in the research due to closures of education settings as a result of COVID-19 pandemic nationwide lockdowns [52]. Table 1 provides information on the demographic characteristics of the participants. Results are presented according to the four overarching themes (see Figure 1). Verbatim quotes from the focus groups are used to support these themes. A visual representation of the data in the form of a word cloud is presented as Figure 2.

### 3.1. Theme 1: Capturing Contextual Information Is Important to Parents and Carers

The first theme relates to the importance of being able to report context when using any type of measurement tool, including situational context (e.g., if a child is unwell), the ability for tools to capture different routines, and some considerations when using accelerometers as lone tools for 24 h movement measurement.

#### 3.1.1. Situational Context

All groups suggested that reporting key contextual information, regardless of measurement tool, was important for measuring the movement behaviours of their young children. Parents specified that it would be important to report on whether their child has a disability, if they suffer from any sleep disturbances or whether they were suffering from an illness at the time of measurement. Parents stated their children’s behaviours could be starkly different if they are unwell, in that they are more likely to be sedentary, nap during the day and run around less when they are unwell. Reporting on illness was therefore important to provide an adequate representation of their child’s usual 24 h day.

‘*I think it all depends on whether they’re poorly or not…[name] sleeps like a trooper when she’s poorly during the day…But then they have their off-days, so it’s like if they’re poorly they’re going to have a nap*.’(P5)

#### 3.1.2. Capturing Different Routines

Parents reported the importance of tools capturing different routines; for example, parents stated that activity would vary depending on if their child was at home or school, at their grandparents or relatives, or if it was a weekday or weekend. Parents were conscious that they do not see their child’s school activity and so they would need to ask the teacher to be able to report this. Parents felt that it was important this information was captured to present an adequate reflection of their child’s activity. For proxy reported tools, it was suggested that a format where there was space for both home and school activities would make it slightly easier to obtain information about school activities. This was also discussed by a nursery teacher in the focus group, who suggested that if proxy reported tools were separated by both school and home activities, then nursery could help to complete the day time sections when children are in school: ‘*We could hand it down from nursery to teatime club or start with breakfast club and then just hand it round*‘ (P6). This emphasised the importance of researchers needing to engage with settings to have teacher involvement.

Additionally, parents reported that activities and sleeping routines are often different on the weekend than the weekdays where the days are structured around school and work. ‘*Yeah. That’s a weekday, but then during the weekend it’s-*’ (P4), ‘*It’s a bit different like us*.’ (P2) ‘*we’re out or we’re going for a walk or we’re at home or -the weekend activity and the week,* yeah.’ (P4) (Focus Group 1 Extract).

Parents suggested that they would like to be able to say why routines and behaviours may be different on certain days and that it would be important to explain any anomalies or provide an explanation. Parents said that having the space for ‘*additional comments’* or to provide more information where you cannot fit it in the generic tool would be helpful to explain anything in more detail. This was thought to be particularly important to explain their child’s sleep characteristics, such as why sleep may be different on certain nights, or provide extra information about the context of sleeping arrangements (e.g., if their child moved into their parents bed during the night).

‘*I definitely think that a box or a question in relation to how often is this usually the case, like being allowed to give a reason as to why on this particular day your child didn’t sleep very long or didn’t actually go to bed because they might have been ill or it might be a weekend, so they might be allowed to have a later night because it’s Friday night*.’(P11)

#### 3.1.3. Using Multiple Measurement Tools Simultaneously

Parents had concerns about the accelerometers being used as a lone tool, particularly for the measurement of sleep, as they thought in these instances it may look as though their child had not slept by examining accelerometer data alone. Parents discussed that their children move around during the night even when they are asleep, either through sleepwalking, having night terrors, or that they move themselves around the bed whilst asleep. Parents proposed that both diaries and accelerometers could be used simultaneously to ensure higher accuracy of their child’s behaviour and so information obtained from accelerometers was not misunderstood: ‘*Then at least you know whether they sleepwalk or they climb mountains or do whatever they do.*’ (P5)

### 3.2. Theme 2: Device Based Tools: Practical Issues

The second theme concerns practicalities of using devices to measure movement in pre-school children, including the comfort, placement, removal and durability of devices. A sub-theme also covers the acceptability of accompanying instructions and logs to report wear time of devices.

#### 3.2.1. Placement of Devices

When asked for initial thoughts on the various devices (Actigraph GT3X+, Actical and activPAL4 micro), participants in all groups showed a clear aversion to devices being attached directly to the skin and most frequently regarded the activPAL as the least accepted tool for these reasons. Participants felt that the device would cause irritation, be painful when removing, were concerned of allergic reactions, described that their child may try pull it off (and subsequently easier to become lost/misplaced) and felt that it would be the most difficult of the devices to re-apply. This was the only device that some of the parents explicitly stated their child would not wear.

‘*Yeah, they wouldn’t want something stuck to their body like that. I just don’t think they’d like it at all. Plasters don’t stay on very well, let alone…*’(P2)

Parents felt that for children with skin conditions such as eczema these types of devices would not be appropriate or feasible: ‘*my son has eczema so he couldn’t have the one that sticks*’ (P11). This same concern was raised in relation to children with sensory issues, as exemplified in the extract below.

‘*I think they’re great. The only issue I have with the actual monitors themselves is how children would react if they have sensory issues. So some children with sensory issues are fine with things touching their skin, but something that’s actually attached to the skin might be kind of irritating for some children*.’ (P9)

Parents reported that devices that could be clipped on or worn on top of clothes rather than being directly attached to the skin were preferred; however, there was variation in preferences for device placement. Some parents preferred the hip placement for devices, reporting that devices would be better located out of sight of the children to reduce the chances that they would play with the device: ‘…*I think if it’s on a belt and you can put like a jumper over the top or something, they’ll forget it’s there*’ (P7). Others stated a preference for the wrist placement as children may think it was a watch, which they reported that children begin to become interested in at this age.

#### 3.2.2. Comfort/Ease of Wearing

There was a lot of emphasis placed on the size of devices, with parents and carers stating that smaller devices would be favoured for this age group. Despite the activPAL receiving negative reactions due to the attachment to the skin, the size of this device was seen favourably. Larger devices may be too big for children of this age, with the Actigraph frequently referred to as ‘*bulky*’ throughout the focus group discussions: ‘*No, I think it’s too bulky for her for it to be on the wrist, she’s only dinky, so I think it [Actigraph] would be too big for her*.’ (P5). However, it was also suggested that there were positives to having larger devices as it would be easier to identify if it had fallen off: ‘…*they’ll not know it’s gone, whereas the ones with the straps you’ll be able to see if it’s going to fall off, it’ll be dangling*.’ (P9).

There were some evident issues raised around the applicability of a child wearing the same device for 24 h per day. This was primarily when the size of devices became problematic. Parents were concerned that a larger device may be uncomfortable when their child was sleeping and may result in the child removing and displacing the device, whilst smaller devices attached to the skin may stay on during the night but would be the least practical in terms of applying and for daytime use. The Actical was thought to be more appropriate for 24 h of wear time: ‘*it’s slimline they might not know it’s there, they might forget*’ (P6) and ‘*Even when they’re asleep it probably won’t dig in as well*.’ (P7). This raised the idea that device preference may also be partly dependent on the behaviour and outcome of interest. For example, larger devices not attached to the skin may be preferable for use during the day but may cause more difficulties during sleep. ‘*But I don’t think this would be worn 24 h. For sleeping, I don’t think it’d [Actigraph] be comfortable, if you turn and it’s on your hip, I don’t think it would be*’ (P2). Smaller devices may be better when only measuring sleep: ‘*Oh, absolutely, because it’s like thinner it would be good for bedtimes and they might not feel it as much, but the other one I don’t think that would be very good for bed*.’ (P7)

#### 3.2.3. Removal of Devices

Parents spoke about the likelihood that children will try and remove the device regardless of where it is placed, as the devices would initially be a ‘*novelty*’ but then children may get bored of them after a while. Parents spoke of this in relation to watches that their children had previously worn: ‘*I just, they’ve worn like little kiddie watches and things before or bracelets, but they always take it, oh, I don’t want to wear that anymore, and they’ll take it off*.’ (P2) ‘*On/off, on/off*.’ (P4) ‘*Yeah, on and off*.’ (P1) (Focus Group 1 extract). Parents did state that regardless of where the device was, even if it was attached in the middle of their child’s back, they would still find a way to take it off if they wanted: ‘*Between the shoulder blades… She can’t reach that… She’ll be like a bear and just scratch it off on the wall*!’ (P2). The groups suggested that if the device was out of sight it may help to reduce the chances of children removing it. However, they also stated that it would be preferable to place the device in a location where children could remove it themselves if they were experiencing any discomfort, rather than something that was attached to the skin that could be painful to remove.

‘*…just for practicality of, just so they’re not stressed as well, if they do want to take it off they can quite easily take it off themselves*.’ (P2)

Although parents thought it was highly likely that the child would want to remove the device at some point, they suggested that the Actigraph and Actical would be easier to reattach as opposed to the activPAL that would require wiping the area and re-applying with new adhesive. This was also a preference for when children are with grandparents or in the care of other adults.

‘*I think the strap one would probably be the easiest for like grandparents and that, because then they don’t have to strip them to stick it back on their thigh*.’(P10)

#### 3.2.4. Durability of Devices

Parents and carers talked about the practicalities of the devices in relation to activities that children engage in stating that devices must be ‘*sand resistant*’ and ‘*water resistant*’. Parents and carers spoke of the need for these considerations when thinking of devices that are wrist worn: ‘*they’re into mud and dirt and soil and digging in the mud*…*And anything else, slime, if we have slime in the water tank or anything like that*’ (P6). Additionally, parents and carers stated the need for devices to be durable and to be child appropriate ‘*childproof, like hard, like not feeble, if you know what I mean-so they’ll easily break*.’ (P7) ‘*Absolutely, no small parts either we have to say these days*.’ (P8) *‘Yeah, very good. Childproof*!’ (P7) (Focus Group 2 Extract).

#### 3.2.5. Acceptability of Accompanying Information—Instructions and Accelerometer Logs

Basic instructions usually administered with an accelerometer (see Appendix A) were well received across participants and were stated to be ‘*straightforward*’ enough to manage the accelerometer wearing protocol. Parents felt that the instructions would be understandable even if children were in the care of family members who did not often use technology. ‘*My grandma’s not good with tech and I think she could understand that, so!*’ (P11).

Participants suggested that an accompanying visual method of instruction would also be useful, such as a link to a video demonstration on how to put the device on, particularly if devices are being used via remote administration, but also to act as a reminder if the device is removed during the week.

‘*Ideally probably a demonstration*.’(P6)

‘*Yeah, that’d be good, like a video or something to go on, wouldn’t it? That would be handy…Like just say oh if you go onto this YouTube website or whatever, it shows you how to do it, because all right it’s reading something, but sometimes I need showing…something to show me what to do. Because then otherwise you might not put it on properly and then it might not get the right reading for what you need*.’(P7) (Extract from Focus Group 2)

Three types of accelerometer log that ranged in the amount of content captured were presented in the demonstration and focus group. Participants across groups consistently reported that they preferred the detailed logs with space for commenting on whether the accelerometer had been worn, if it had been removed, time and reason for removal, and information on bed and wake up times. Parents reported that this format made it ‘*easier to figure out*’ what needed to be reported and how to keep track of when the child was wearing the accelerometer. Simple logs containing less information (e.g., just the time monitor was removed, time monitor was put back on, and reason) were seen to be more complex in that parents would have to think more about what they had to write: ‘*It looks a little bit like…Like where to start, am I putting the right thing in*.’ (P4).

Parents stated that plenty of space would be needed on the accompanying accelerometer logs to state when the device had been removed, so that even if children did remove the device frequently, they were able to report this*:* ‘*Yeah, I just think I need more room for how many times it would be removed*!’ (P5)

### 3.3. Theme 3: Encouraging Children to Wear a Device

The third theme concerns encouraging children to wear the devices, including compliance and ideas on how to make tools more appealing to children of this age.

It was apparent through the focus group discussions that parents and carers thought that compliance with the accelerometer protocol was child dependent. Some suggested that they would not have any real problems with their children wearing any of the devices: ‘*Yeah, that’s the only thing. If I didn’t have a choice I wouldn’t be too bothered. So I would potentially use any and I think she’d do any…*’ (P4). Others felt they would really struggle to get their child to comply with the device wearing. ‘*I don’t think mine would wear them. I don’t think. This one I think would be a novelty to start with*.’ (P2). Although parents offered some practical solutions to try and increase compliance, it may be that to a certain extent it is individualistic as to whether children would take part, demonstrating the importance of over-sampling in research studies with pre-school children: ‘*…every child’s different. Some children might leave it alone, they might adapt to it brilliantly; whereas, other children might be like no they just don’t like wearing it*.’ (P8)

Participants confidently reported that the way in which devices are framed to children could impact on compliance ‘*Just because they look cool, they’re kind of fun and if it’s explained to them in a way they can understand they might think it’s really cool*’ (P11). An example that arose through all focus groups was framing the device to children as something that could ‘*give you superpowers’* or that was a ‘*magic belt*’. During one focus group, a child showed interest in the devices, so their parent asked them if they would want to wear the device to get superpowers, in which the parent stated: ‘*[Child] just said he wants superpowers, so he’d have it on all day long*.’ (P9). It was suggested that devices could be turned into a game, in particular, this was stated in relation to the Actigraph: ‘*So, I like the Actigraph one, although it’s chunky and everything, I don’t think that would be an issue with kids because it stands out for them and you can make it a game*.’ (P9).

Similarly, suggestions were made on how to make the devices more appealing for children so that they would be easier to use, including the use of colourful and different patterned belts ‘*…if they were more appealing, more child friendly. Like colourful bands if you want them to wear it and like you know with like the characters and things*.’ (P2). Further suggestions included being able to personalise the devices in some way, such as children designing stickers or pictures that they could put on their device: ‘*Create your own one. Give the kid some stickers to decorate their own*’ (P7). Alongside making the device more appealing to children, these strategies would help practically, to determine who the device belongs to if it falls off whilst at nursery: ‘*And you would know who’s was who, especially around nursery, because then you could just say… this child’s has got the flower on, this one’s got a car on. So then you’d be like right oh you’ve lost yours, there you go, rather than going like this is number 574, whose is this!*’ (P8).

Parents stated that for children with disabilities, if some of the devices (that did not have a direct skin attachment) were framed appropriately or explained to the child then they could be applicable. With this, it was evident that there needed to be a degree of parental buy-in to encourage children to wear the devices and to help facilitate compliance from their children.

‘*I’ve got a son that’s got these special boots. Like my kids know that he’s got to wear them. So I reckon if I told them they had to wear that they’d just wear it*.’ (P2)

### 3.4. Theme 4: Presentation of Diaries and Questionnaires

Theme four is specific to the presentation of diaries and questionnaires, including the format of these types of tools, timing of data collection and ensuring age specific activities.

#### 3.4.1. Examples of Movement Behaviours

Ideas about the presentation of the proxy reported tools included that it was favourable if tools were clearly aimed at young children. One parent reported that having examples of activities that your child would ‘*typically actually do*’ made it more appealing than something that appeared to have unrealistic expectations of young children and the fact that children ‘*don’t sit still and play nicely*’ (P11). Parents also suggested that as children engage in so many activities, and that these vary per day, it would be useful to have space to report the different activities that their child does, without always having to stay within prescribed activities listed. However, it was recommended that tools should contain some age-specific activities, so that parents can understand the aim of the question but with additional space for reporting further activities.

‘*Yeah. So we could just have examples so people don’t think hmm, what have they done?… Yeah, but then have the choice to also write your own if you can do that*.’(P4)

#### 3.4.2. Appearance of Diaries and Questionnaires

There was some contradiction between groups as to whether the proxy reported tools should be made visually appealing or made to look ‘*boring’*. Parents reported that the tool could be quite intimidating and daunting, but if it was colourful or was made to look appealing in some way then it would be more likely that they would want to complete it: ‘*And maybe colour coordinate it… just something that makes it look a bit like that other thing that you’ve got…Yeah, just pleasing to the eye and like you, I don’t know, it sounds silly, but you want to fill it out, because it’s… it looks more pleasing, doesn’t it*…’ (P4). Others suggested that it would be preferable for the tool to be as dull as possible, as otherwise their children may think that it is for them: ‘*Exactly, then they’d be like oh I’m just going to colour that in for you mummy, that’s all right. It’s like, don’t do that! But that’s what kids do-any bit of paper they find they’re like oh I’m just going to draw mummy a nice picture on that. They don’t think it’s important.*’ (P8).

#### 3.4.3. Format of Tools: Paper or App?

Whilst there was little discussion around the mode of administration, in one focus group, it was suggested that a phone application (app) rather than a paper diary may be helpful, particularly for sleep. It was suggested that normally parents will pick up their phone to see what time it is when their child wakes up during the night. ‘*Yeah, or even like you say yeah an app would be good*…*An app would be easier, absolutely*… *you could literally tick a box, bedtime… well, they got up at this time, and then it’s so much easier keeping track rather than… having to write this down*.’ (P8)

#### 3.4.4. Recall or Real Time

Parents reported that questionnaires with previous week recall were confusing and too difficult to complete. Recall was a concern that continued to arise throughout the focus groups, both in terms of how parents felt they would not accurately retrospectively report past week activity, but also their concern that they do not always know if children have woken up during the night so this would not be correctly detected:

’*I think it’s a bit too hard for me this, because like she does so much each day, like all the different things she does through the day, and then you’re having to record them and remember, oh, it’s so confusing, I just don’t understand it*.’(P3)

Rather, parents suggested that having space to report every day would be much more beneficial: ‘*Most of this it would. I personally would find it easier having it daily instead of how many days this week*’ (P5). Parents felt this would be easier as they would not have to think back to the past week and what their child spent the whole of each day doing, were less likely to get children mixed up, but would also be able to capture the intricacies and differences between each of the days. In this regard, the proposed sleep diary was generally well accepted and understandable, with participants suggesting that filling in the parental reported sleep diary for a seven day period was ‘*Oh, easy*’ and would ‘b*e straightforward*’. A diary-based format was preferable over a questionnaire; parents stated that a questionnaire may take less time to complete, but they felt that having the space to report in more detail and have a longer tool would be beneficial in terms of the tool meaningfully detecting the activity of their child.

‘*It might mean it’s more paperwork, but it makes it more- easier to look at. So it kind of looks like it’s more but it’s…*’ (P4) ‘*In the long run it’ll be easier.*’ (P2) ‘*I think it’s going to be easier, especially if you’ve got more kids, because then you’re not getting the kids mixed up either, because I know that’s what I do. I’m like, [child’s] done this today, no, he hasn’t, it was [other child], no, it was [other child]’*.(P5), (Focus Group 1 Extract)

## 4. Discussion

### 4.1. Main Findings

The results of this study revealed four main themes on the feasibility and acceptability of measurement tools used to assess movement behaviours of pre-school children from the perspective of parents and carers. These include:Importance of providing contextual information when using any measurement tool to report on children’s movement behaviours (e.g., child illness, capturing different routines such as home and school activity, and potentially using multiple measurement tools to obtain context alongside device based tools).Practical issues associated with devices (e.g., placement of devices and aversion to devices being attached directly to the skin of their child; concern of larger devices during sleep time; likelihood of removal of devices (and subsequently needing space to report this on an accompanying log); worry about durability of devices during child play; and preference for written and visual instructions for device administration).Encouraging children to wear a device (e.g., making devices attractive to children-‘superpowers’, colourful belts and personalised stickers).Presentation of diaries and questionnaires (e.g., parents would prefer examples of age appropriate movement activities, much prefer real-time recording than recall).

Overall, the activPAL was the least preferred device and raised the highest amounts of concerns for children in this age group, namely due to this tool being attached to the skin. This finding is in line with previous research reporting that pre-school children ‘strongly opposed’ to having chest worn devices attached to their skin including requesting to remove the device due to it feeling uncomfortable and slipping from their skin during a physical activity session [29]. Similar concerns have been raised with the use of the activPAL due to skin irritation in previous research [53]. However, further studies have reported that the activPAL was generally accepted [24,28], with minimal reports of removal of the device due to the dressing of the device being uncomfortable for the child [24]. Despite this, in some research studies it may be that, for accuracy reasons, devices must be attached to the skin. In these instances, researchers may want to address some of the perceived concerns including a demonstration of the device attachment and removal both to children and the caregivers, and clearly explain what the device is for in an interactive and fun way to children prior to administration.

The Actical was the preferred device overall and raised fewest issues for use with young children, mainly due to the smaller size of the device and that attachment was not directly to the skin. In a previous study, researcher reported acceptability of the Actical and activPAL suggested that most children had high acceptability to wearing both devices. However, children only wore the devices for one day in this study and authors noted that a proportion of children did not assent to wearing the devices and some were lost/removed during the day [26]. Where possible, devices similar in size to the Actical may be preferred for measuring movement behaviours in this age group.

The Actigraph was generally well accepted by parents and carers, however there were some concerns raised with the size of this device mainly for sleep, with parents stating that the device was ‘bulky’ and may be uncomfortable if a child rolls on their side whilst asleep. However, it was also suggested that the size of this device made it less likely to become misplaced. Previous research has highlighted that the Actigraph accelerometer is generally well accepted by young children [25], with few reports that the accelerometer sometimes moved out of the correct position [54]. It is important to note that these previous studies have focused on day time activity rather than sleep. Removal of device based measurement tools is a long standing challenge with measurement of physical activity and related behaviours of young children [36]. Whilst it may be inevitable that devices are removed for at least part of the measurement period, ways to make the tools more attractive and fun may help to increase compliance with the accelerometer wearing protocol. With this, parents suggested that the Actigraph would be the easiest device to turn into a game and if framed appropriately in a fun way, and with children having autonomy to make the device their own, then children may enjoy wearing the device. Previous research reported that children enjoyed wearing the Actigraph accelerometer by thinking of themselves as some of their favourite characters by wearing the device [55]. Parents and carers suggested that the use of stickers and colourful belts would make the devices more appealing to children, a strategy that has been used in a previous piece of research to try and increase willingness and compliance with pre-schoolers to wear devices [26]. This may be a helpful consideration in the design and development of device-based tools and their wear protocol.

Actigraph and Actical accelerometers have most frequently been validated and worn using a hip placement [56,57,58]; however, both hip and wrist placements have shown potential to classify intensity of movement in pre-school children [47]. The findings of the present study showed variation in terms of preference between wrist and hip; with wrist being said to be feasible if the device can be worn like a watch, and the hip placement if the device is under the clothes then children are more likely to forget about it. If tools are framed in an attractive way to children as recommended by parents, placement location may be less relevant or problematic. In light of this, the most accurate placement for the device may be chosen with likely minimal impact on the acceptability of the tools.

In previous physical activity intervention studies using the Actigraph accelerometer, it was reported that parents sometimes forgot how to apply the device during the seven day protocol [55]. In the present study, participants reported that written instructions accompanied with video demonstrations would be helpful; this would also help with remote administration and reduce the need for research assistants to visit families during the study. Similarly, parents reported that having detailed accelerometer logs that include bed time and wake time were easier to follow for reporting device wear time and removal, and were less complex than logs with less information.

In relation to their own experiences, participants were very aware of recall bias when responding to questionnaires. Similarly, given their busy schedules with having young children, parents felt that retrospective reporting may be too difficult and confusing. Parents felt diary based tools would be a much easier and more accurate method of measurement as these could be completed daily. Where feasible (in terms of resources including researcher time and financial resources), it may be more appropriate to use diaries rather than questionnaires. These findings highlight the importance of involving the target population in the development of new tools to ensure that the tool will be meaningful to complete, that the items on the tool capture the appropriate and relevant behaviours, and that the questions are understandable [59]. Additionally, parents suggested that having space to help put their responses or data from accelerometry into context, as well as being able to explain any anomalies in their child’s behaviour, was important. Not only would this help with contextualising the information but may also help parents feel their children’s behaviours are more accurately captured. These findings further demonstrate the importance of public involvement and engagement in the design and conduct of research, to ensure that subsequent participation in research is meaningful through appropriately co-designed study materials [60].

Further to this, it was suggested that the use of electronic formats of data collection, such as phone apps, may provide a welcome alternative to traditional paper formats of questionnaires and diaries. Such methods are already seen in national data collection schemes such as the Active Lives Survey, where both electronic and paper based formats of the questionnaire are available [61]; however, this does not include proxy reporting for children as young as 3 and 4 years old. Compliance issues have been reported for electronic surveys compared with telephone surveys, with particularly high attrition rates for parents with young children [62]. As such, it is important that electronic-based methods of data collection for the measurement of movement behaviours of young children are thoroughly evaluated to ensure accuracy and feasibility prior to use [62].

### 4.2. Recommendations

Based on this research, we have provided a series of recommendations for the measurement of movement behaviours of pre-school children that may be used in the design, development and implementation of research projects (see Table 2, can also be found in the form of an infographic in Appendix A).

### 4.3. Strengths and Limitations

To our knowledge, this is the first study to assess the acceptability and feasibility of a range of measurement tools used to examine the movement behaviours of pre-school children from the perspective of parents and carers [22,23]. This study used a novel methodology for the measurement field that is largely based on quantitative studies only. Through this, the study has highlighted practical issues and helpful insights that researchers may wish to consider when designing studies and selecting materials to use for measuring the movement behaviours of pre-school children, for example for intervention evaluation [63]. Moreover, the recruitment strategy was used to deliberately draw from groups whose views are often underrepresented in measurement literature [22,23]. Although this study included a small sample size, the study offers practical insights and strategies into the design of studies examining movement behaviours of pre-school children that may be generalisable through transferability; however, we will leave this at the discretion of the reader as to whether the findings apply to their group of interest [64]. The insights obtained by the parents and carers involved in this research could be shared with populations of interest through public involvement and engagement prior to full scale studies.

This study also has some limitations that must be recognised. Firstly, despite trying to achieve a diverse sample by recruitment of two schools in one of the most ethnically diverse cities in the UK [65], the sample in the present study was not ethnically diverse; as such, the generalisability of the findings across different ethnic groups is unknown. In the future, working with established gatekeepers in the early years settings, with whom individuals of different ethnicities identify with and trust may help to recruit a more diverse sample. Similarly, although an advantage of this research was the recruitment of underrepresented groups, it would be useful to tests these tools using similar methodology across a range of socioeconomic status groups. Although participants had the opportunity to trial the tools during the demonstration conducted as the first part of the session, they did not use the measurement tools for extended periods of time prior to the focus groups. It is plausible that in practice more feasibility concerns could be raised than those discussed in this research. The aim of this research was to explore the acceptability and feasibility of the tools prior to use in future research projects. Therefore, although this study provides important insights into the feasibility, acceptability, and practicality of a range of measurement tools that may otherwise not have been considered, we cannot determine from this study whether or how this would impact compliance in research studies. Finally, this research was conducted prior to the coronavirus pandemic, it is therefore plausible that some of the issues raised may be exacerbated as a result of this. For example, participants may be more vigilant to the cleanliness of device-based tools.

In this study, we presented certain models of devices but due to rapid and continuous advances in accelerometry technology, it is likely that newer versions of the devices will be available, now and in the future. The findings of this research are not intended to be a criticism of any particular devices but rather to present practicalities of using the devices from the perspective of parents of young children, to help with the development and implementation of research studies with this age group. We hope that the findings of this study will also be useful to manufacturers in future development of revised versions of the tools.

### 4.4. Implications and Areas for Future Research

This research provides a series of recommendations for the measurement of movement behaviours of pre-school children that may be used in the design, development and implementation of future research projects. These recommendations provide an important contribution to the literature, given the lack of information on the acceptability and feasibility of measurement tools in this age group, and are particularly valuable given that the recommendations are drawn directly from the views of parents and carers of young children. Further to this, the insights from parents and carers can help with appropriate development of new tools.

The work also highlights some important areas for future research. Firstly, although a range of tools were assessed in the research, there are a wide variety of tools including a plethora of different device based tools, and new devices continuously becoming available. There is a clear need for more research assessing the acceptability and feasibility of different measurement tools using qualitative research methods. In particular, there is a need to integrate evaluation of acceptability and feasibility in studies that examine validity and reliability of measurement tools, to demonstrate the full picture on the usability of the tool. In line with this, the work presented here can be used to inform further validity and reliability studies, with feasibility and acceptability being as important as other measurement properties for the success of a study.

## 5. Conclusions

This novel study highlights the importance of being able to provide contextual information when reporting movement behaviours of young children (e.g., child illness, capturing different routines); some practical issues associated with devices (e.g., aversion to devices being attached directly to the skin of their child; concern of larger devices during sleep time); ways to encourage children to wear a device (e.g., making devices attractive to children- ‘superpowers’, colourful bands and stickers); and preferences on presentation of diaries and questionnaires (e.g., age appropriate activities, preference for real-time recording over recall). The findings of this research can be used to help with the design, development, and implementation of studies measuring movement behaviours of pre-school children.

## Figures and Tables

**Figure 1 ijerph-19-03733-f001:**
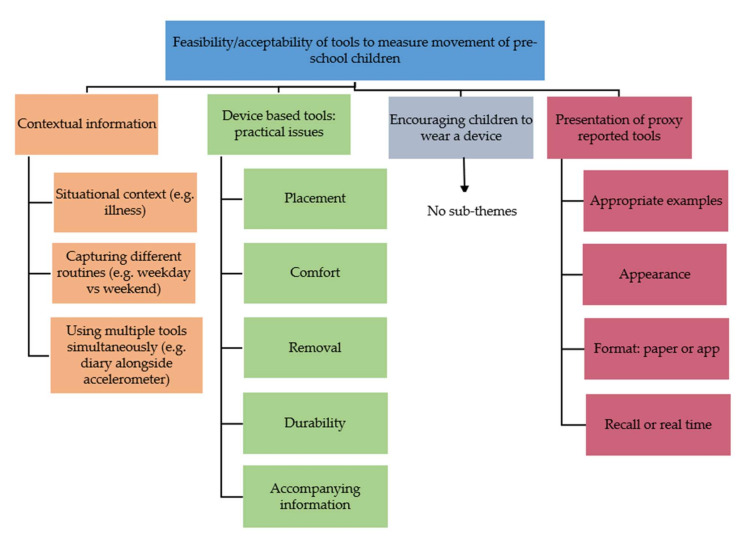
Overview of key themes and subthemes.

**Figure 2 ijerph-19-03733-f002:**
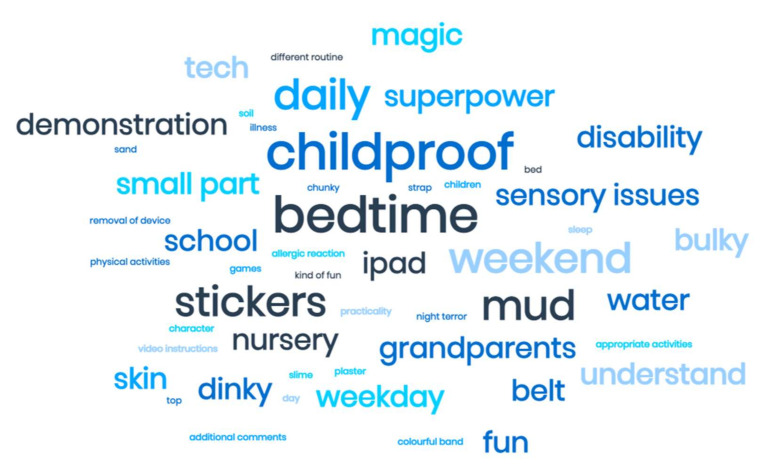
Word cloud representation of the data.

**Table 1 ijerph-19-03733-t001:** Demographic characteristics of participants.

Demographic Characteristics		*n* = 11
Sex (%)	Female	100
Ethnicity (%)	White British	100
Age of parent/carer (years)	Median	29
	Range	21–61
Age of pre-school child participant cared for (years)	Median	3.7
	Range	3.3–4.9
Education level (%)	Masters/PhD or equivalent	0
	Bachelor degree or equivalent	9
	A levels or equivalent	9
	Diploma in higher education/BTEC or equivalent	18
	GCSE’s or equivalent	27
	Vocational qualifications (NVQ Level 2)	9
	National nursery examination board	9
	No formal qualifications	9
	Did not specify	9
Employment status (%)	Working full-time	36
	Working part-time	27
	Looking after the home	9
	Not working	27
Household income per year (%)	<£4999	0
	£5000–£9999	9
	£10,000–£14,999	9
	£15,000–£19,999	18
	£20,000–£24,999	18
	£25,000–£29,999	18
	£30,000–£34,999	0
	£35,000–£39,999	9
	>£40,000–£44,999	0
	Don’t know	18
Index Multiple Deprivation quintile (%)	1	81
(1 = most deprived, 5 = least deprived)	2	9
	3	0
	4	0
	5	9

**Table 2 ijerph-19-03733-t002:** Recommendations for measuring movement behaviour of pre-school children.

Recommendations and Practical Considerations	
1	Context is important to parents and carers—having space to explain their child’s health status (e.g., illness) helps make measurement meaningful.
2	Ensuring measurement captures different routines to be reflective of children’s movement e.g., home vs school, weekday vs weekend.
3	Devices worn as watches or placed out of sight were preferred. Devices stuck to the skin were less favourable.
4	Smaller devices preferred for 24 hour movement measurement, but different devices may be favoured if measuring only one proportion of the 24 hour day (e.g., larger devices for day time, smaller devices for night time).
5	Removal of devices by young children may be inevitable so include detailed device wear time logs with plenty of space to report when the device is worn/removed.
6	‘Child-friendly’ devices—suitable for playing in mud, slime, sand and water—and no small parts! Demonstrate safety of devices prior to studies.
7	Written and visual (video demonstrations) study instructions are most helpful to act as a reminder during the measurement period.
8	Frame research to young children so that taking part is ‘cool’—devices can give ‘superpowers’ or ‘magic’.
9	Modify tools to make them ‘childlike’ e.g., so children can personalise their device with stickers or provide devices with colourful belts or with children’s favourite characters on.
10	Daily reporting easier for proxy reported tools—recall can be particularly challenging with young children. Ensure that age appropriate activities are included on the tool.

## Data Availability

Not applicable.

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
