# Peer review of "Parental Views on the Acceptability and Feasibility of Measurement Tools Used to Assess Movement Behaviour of Pre-School Children: A Qualitative Study"

_ijerph, 2022, doi:10.3390/ijerph19063733_

Round 1

Reviewer 1 Report

I think this study is important and novel, and I very much enjoyed reading it (both as a physical activity researcher and as a parent). I don’t really have any major comments to make.

The main issues that I wondered about were addressed in the discussion: the representation in terms of ethnicity, and the short duration of time that these devices were encountered by the parents. For the former point, can the authors perhaps illustrate why they encountered difficulty with recruitment across different ethnic groups, and how these might be overcome in future? With respect the latter point, I think that this study still gives important information that would impact on recruitment to accelerometer studies in pre-school children.

This is a very interesting and engaging piece of work, and I wish the authors the best with its publication.

Author Response

Dear Reviewer,

Thank you very much for taking the time to consider and review our manuscript, and for your helpful comments and feedback. This is a very interesting point about overcoming the lack of ethnic diversity within our sample. In response to this, we have added the following sentence to the revised manuscript: ‘In the future, working with established gatekeepers in the early years settings, with whom individuals of different ethnicities identify with and trust may help to recruit a more diverse sample.’ (Lines 652 to 655).

Many thanks.

Reviewer 2 Report

The abstract and the keywords are clear enough and follow the contents of the paper. Overall, the paper presents a logical order of ideas about an interesting topic. The introduction is organized from the general to the particular, culminating in the formulation of objectives. The author presents a critical and sufficiently broad review of the existing theoretical-empirical evidence. The state of the art supports the problem. Limitations were adequately pointed out in the paper.

The quality of the written and readability of the paper in English is fine. Needs punctuation improvement.

The sample and sample selection are adequately described. However, the sampling process and its representativeness are reported as a limitation. The material and methods of data collection are adequate and are clearly described. The methodological design allows to investigate/test of the formulated objectives. The quality of the methods used is guaranteed, and the ethical principles are safeguarded. The data analysis options are adequate and properly described.

The results seem credible and consistent with the title and objectives. The results are properly presented and the presentation of data and results is clear. Tables and figures are well-designed, relevant, and self-readable, allowing for easy reading and analysis.

The paper brings to the debate the acceptability and feasibility of a range of measurement tools used to assess the movement behaviours of pre-school children, in parents and carers living in areas of high deprivation. Despite the range of limitations listed by the authors, the discussion is pertinent, supported by the state of the art. The results and their implications/applications should be discussed in the broader context.

The conclusions follow from the presented results and systematize the central ideas of the work. The findings are presented as conclusions. We suggest that future research and lines of research should be better listed and highlighted.

The references are relevant to the text.

Author Response

Dear Reviewer,

Thank you very much for taking the time to consider and review our manuscript. We are very grateful for the helpful comments and feedback you have provided. In response to your comments, we have gone through the manuscript to improve the punctuation. In response to the latter comments, we have added a new section to the revised manuscript ‘Implications and areas for future research’ (Lines 677 to 695), which reads as follows:

‘4.4 Implications and areas for future research

This research provides a series of recommendations for the measurement of movement behaviours of pre-school aged children that may be used in the design, development, and implementation of future research projects. These recommendations provide an important contribution to the literature, given the lack of information on the acceptability and feasibility of measurement tools in this age group, and are particularly valuable given that the recommendations are drawn directly from the views of parents and carers of young children. Further to this, the insights from parents and carers can help with appropriate development of new tools.

The work also highlights some important areas for future research. Firstly, although a range of tools were assessed in this research, there are a wide variety of tools including a plethora of different device based tools, and new devices continuously becoming available. There is a clear need for more research assessing the acceptability and feasibility of different measurement tools using qualitative research methods. In particular, there is a need to integrate evaluation of acceptability and feasibility in studies that examine validity and reliability of measurement tools, to demonstrate the full picture on the usability of a tool. In line with this, the work presented here can be used to inform further validity and reliability studies, with feasibility and acceptability of tools being as important as other measurement properties for the success of a study.

We also refer to these wider implications through the discussion, including in the ‘Strengths and Limitations’ section, lines 635 to 640, which reads as: This study used novel methodology for the measurement field that is largely based on quantitative studies only. Through this, the study has highlighted practical issues and helpful insights that researchers may wish to consider when designing studies and selecting materials to use for measuring the movement behaviours of pre-school children, for example for intervention evaluation [63].’

Many thanks.

Reviewer 3 Report

Manuscript ID: ijerph-1624116

The Authors of Parental Views on the Acceptability and Feasibility of Measurement Tools used to Assess Movement Behaviour of Pre-school Children: A Qualitative Study provide an interesting and novel insight into the knowledge about the equipment and tools used in the research of physical activity in preschool children, especially in deprived areas.

The paper is well composed, thoughtful, easy to read and very interesting. I recommend some minor improvements as follows:

  1. Include the SUNRISE study results description into the Introduction section Publications – SUNRISE (sunrise-study.com) as an international project covered the analysed age group and used various measurement tools to assess the movement behaviours.
  2. Relocate the detailed information about:
  • number of focus groups
  • number of participants in focus groups
  • demographic information about participants (parent/kids)

into the 2.1. Participants section.

  1. Consider enriching the description of the results by including one of the methods of visual presentation of qualitative data, e.g.: coding stripes, word cloud, word tree or explore diagram.

I recommend the manuscript ijerph-1624116 to publish in IJERPH after the minor revision.

Kind regards,

Reviewer

Author Response

Dear Reviewer,

Thank you very much for taking the time to consider and review our manuscript. We are very grateful for the helpful comments and feedback you have provided. Please see below where we outline how we have addressed each of these comments:

  1. We have added the following information to the Introduction, lines 66 to 69, regarding the SUNRISE study results:

‘For example, the SUNRISE study, an international project using multiple measurement tools to collect data on movement behaviours of pre-school children, has suspended use of an accelerometer and modified existing parental reported tools due to feasibility and acceptability concerns.’

  1. We have made the decision to keep the information on the participant sample in the Results section, as we believe these are the results of our study, rather than methods that were used. Whilst we outline the sampling and recruitment strategy in methods, we believe it makes sense that the results including how many participants and information about the participants should be displayed in the Results section.
  2. We have developed a word cloud as a method of visual presentation of the qualitative data presented in this research, displayed in the revised manuscript as Figure 2 (lines 498-517). If it is felt that this improves the quality of the paper then this can be included. We also provide a visual presentation of the themes and subthemes in the ‘Figure 1’, which can be found from lines 178 to 203.

Many thanks.